# HLA-DR Expression on Monocytes and Sepsis Index Are Useful in Predicting Sepsis

**DOI:** 10.3390/biomedicines11071836

**Published:** 2023-06-26

**Authors:** Bibiana Quirant-Sánchez, Oriol Plans-Galván, Ester Lucas, Eduard Argudo, Eva María Martinez-Cáceres, Fernando Arméstar

**Affiliations:** 1Immunology Division, LCMN, Germans Trias i Pujol University Hospital and Research Institute, FOCIS Center of Excellence UAB, 08916 Barcelona, Spainevmcaceres@gmail.com (E.M.M.-C.); 2Department of Cell Biology, Physiology and Immunology, Universitat Autònoma de Barcelona, 08193 Bellaterra, Spain; 3Intensive Care Unit, Hospital Germans Trias i Pujol, 08916 Badalona, Spaineduargudo@gmail.com (E.A.); 4Departament of Medicine, Universitat Autònoma de Barcelona, 08193 Bellaterra, Spain

**Keywords:** sepsis, biomarker, sepsis index

## Abstract

The reduction of mortality in patients with sepsis depends on the early identification and treatment of at-risk patients. The aim was to evaluate the HLA-DR expression on the surface of monocytes (_M_HLA-DR ratio), the sepsis index (CD64 expression on neutrophils/_M_HLA-DR ratio), and C-reactive protein (CRP) with the development of sepsis. We prospectively enrolled 77 critically ill patients, 59 with stroke and 18 with traumatic brain injuries. The biomarkers were tested at the baseline and 3, 6, 9, 12, and 15 days later. Most patients (71%) developed sepsis (4.2 ± 1.3 days after admission). On day 3, those subsequently developing sepsis had lower levels of _M_HLA-DR+ (81.7 ± 16.2% vs. 88.5 ± 12.1%, *p* < 0.05) and higher sepsis indexes (0.19 ± 0.19 vs. 0.08 ± 0.08, *p* < 0.01) than those not developing sepsis. The _M_HLA-DR ratio slowly recovered before day 6, while the sepsis index remained raised in septic patients up to day 9 (*p* < 0.05). To predict the development of sepsis, optimal cut-offs were CRP levels > 106.90 mg/mL (74.19% sensitivity, 69.49 specificity) and _M_HLA-DR expression rate < 72.80% (45.31% sensitivity, 89.47% specificity). The periodic monitoring of the _M_HLA-DR expression together with CRP and sepsis index may help to identify patients in the ICU at increased risk of developing sepsis.

## 1. Introduction

Sepsis, particularly septic shock, causes high mortality rates in intensive care unit (ICU) patients [1]. The early identification of at-risk patients may likely help to better tailor therapy in order to decrease the risk of death [2,3,4,5,6].

The research on new biomarkers that allows us a faster detection of sepsis is essential to reduce mortality and morbidity rates. Numerous molecules of immune failure have been studied. Reversible immunodepression (i.e., immunoparalysis) has been associated with increased susceptibility to infections in critically ill patients. Monitoring the immune status of at-risk patients may likely help us recognize immunoparalysis early and, thus, identify patients with an increased risk of infectious complications [7]. On the other hand, other studies have focused on the search for biomarkers that are associated with favorable prognoses. In this context, the study by Cejková et al. demonstrated that increased prolactin mRNA expression in monocytes is associated with better prognoses in hematologic patients. [8]

A number of molecules associated with immune failure have been studied. The best biomarker to monitor immunoparalysis is human leukocyte antigen-DR (HLA-DR) because it has the following characteristics: this molecule and expression is used within the immune system, specifically by monocytes, to present pathogen antigens and the activation of T lymphocytes. Therefore, a lower expression on the surface of monocytes (_M_HLA-DR) is related to a lower capacity of the immune system to respond to an infection [9,10,11]. Moreover, since it is a very early step in the immune response to pathogens, it is a very early biomarker of immunosuppression. HLA-DR has been proposed as a predictor of septic complications in critical conditions by many studies [12,13,14]. Another biomarker to monitor the immune status is the CD64 molecule, which is induced in neutrophils within a few hours after being in contact with bacteria. The increase in CD64 expression on neutrophils (_N_CD64) allows differentiation between resting and activated neutrophils and can be useful as a biomarker for infection monitoring [12,14,15,16,17].

Research on biomarkers that can enable more rapid detection of sepsis is essential to reduce mortality and morbidity rates. In the present study, we propose to investigate the role of HLA-DR expression on monocytes, the sepsis index (ratio between _N_CD64 and _M_HLA-DR), and CRP as predictive biomarkers of sepsis.

## 2. Materials and Methods

### 2.1. Patients

Seventy-seven critical neurological patients admitted to the ICU of the Germans Trias i Pujol University Hospital without infection were included in a longitudinal prospective study over 24 months.

Severe neurological patients were chosen for the study because they are a relatively homogeneous group of patients who are admitted without being infected, whose stay in the ICU is prolonged, and who usually become infected during their admission. The inclusion criteria were as follows: (1) at least 18 years of age and (2) without infection but with serious neurological pathology.

The exclusion criteria were as follows: (1) immunocompromised patients; (2) patients under 18 years of age; and (3) patients who died within 24 h after admission. Patients were monitored daily both clinically and analytically to detect sepsis and were classified as “septic” or “non-septic”, as well as patients who developed septic shock according to the Sepsis-3 definition [18,19]. In addition, the severity index was assessed by calculating the acute physiology and chronic health evaluation (APACHE) II score and the sequential organ failure assessment (SOFA) score upon admission (Table 1). Every patient was monitored for 28 days after admission.

This study was approved by the Ethical Committee of Germans Trias i Pujol Hospital (PI-15-081), and all patients or their relatives gave their informed consent according to the Declaration of Helsinki.

### 2.2. Definitions

Infection was defined as a pathological process caused by the invasion of normally sterile tissues, fluids, or body cavities by pathogenic or potentially pathogenic microorganisms.

Sepsis (based on the Sepsis-3 conference) was defined as a “life threatening organ dysfunction caused by a dysregulated host response to infection”. Furthermore, patients were classified according to the development of septic shock, defined as a “subset of sepsis in which underlying circulatory and cellular/metabolic abnormalities are profound enough to substantially increase mortality”.

Serious neurological pathology was defined as stroke and traumatic brain injury with a decreased level of consciousness that required admission to an ICU.

Immunocompromised patients were defined as patients diagnosed with any type of primary or acquired immunodeficiency (e.g., HIV-positive, immunosuppression, patients exposed to chemotherapy, radiation, or steroids during an extended period of time or at high doses) or with a pathology advanced enough to suppress defenses against infection, e.g., leukemia or lymphoma.

### 2.3. Samples

Samples were obtained in the first 24 h after admission to the ICU, and the analysis was performed within 4 h after blood extraction. Blood extraction for each patient was repeated every 72 h for 15 days unless an early termination was established because of exitus.

### 2.4. Flow Cytometry Staining

Blood samples from the patients were collected with EDTA anticoagulant. An amount of 100 μL of whole blood was used, which was stained with 15 μL of CD64 PE, 5 μL of CD15 APC, 2.5 μL of HLA-DR BV421, 2.5 μL of CD14 APCH7, and 2.5 μL of CD3 BV515 (all from BD Biosciences, San José, CA, USA). After staining, cells were incubated for 20 min in darkness at room temperature (22–24 °C), followed by erythrocyte lysis, performed using BD FACS Lysing Solution (BD Biosciences) for 7 min. After centrifugation (1300 rpm for 5 min) and washes with FACSFlow (BD Biosciences), samples were acquired using a BD FACSCanto II (BD Biosciences). A minimum of 10,000 monocytes were recorded per sample.

### 2.5. Flow Cytometry Calibration

To standardize the analysis, we used Rainbow Calibration Particles (6 peaks; BD Biosciences), which contain a mixture of particles of similar size with different fluorescence intensities. The particles were used according to the manufacturer’s protocol and were reconstituted with phosphate-buffered saline (BD Biosciences). Equipment voltages and the mean fluorescence intensity (MFI) were adjusted for each fluorochrome on a daily basis in order to standardize the protocol, reducing the existing inter-test variability.

Calibration was performed at the start of the experiments (both baseline and follow-ups). All included patients were analyzed with the same voltage setting of the cytometer lasers.

### 2.6. Flow Cytometry Analysis

The MFI of CD64 on neutrophils (_N_CD64) and MFI of HLA-DR on monocytes (_M_HLA-DR) and on lymphocytes (_L_HLA-DR) were measured (Figure 1). In addition, the HLA-DR expression rate on monocytes was measured. For the analysis, we used HLA-DR expression on lymphocytes as negative control and CD64 on monocytes as a positive control. The analysis of cell subpopulations was performed using the FACS Diva version 6.1.2 software (BD Biosciences).

### 2.7. Plasma Analysis

C-reactive protein was analyzed in plasma samples of patients at each time point using AU-5800 (Beckman Coulter, Brea, CA, USA) immunoanalyzers.

### 2.8. Statistical Analysis

Quantitative variables are presented as the mean ± SD and qualitative variables as percentages and numbers. Normality criteria were determined using the Shapiro–Wilk test. The Mann–Whitney *U*-test was used to compare differences in the CRP, lymphocyte count, HLA-DR expression rate, MFI of _M_HLA-DR, HLA-DR index, and sepsis index between outcome groups at each time point. In order to evaluate more stable biomarkers, we assessed the HLA-DR index, which was defined as the ratio between the MFI of _M_HLA-DR and _L_HLA-DR. Moreover, we analyzed the sepsis index, defined as the ratio between _N_CD64 and _M_HLA-DR, as previously reported by other authors.

Analyses of the HLA-DR index and sepsis index were performed after logarithmic transformation, given that the distribution was evaluated as log-normal. Unadjusted and adjusted linear regressions were used to evaluate _M_HLA-DR in relation to different time points before and after infection. Linear mixed models for repeated measurements were used to evaluate the dynamic variation in _M_HLA-DR and sepsis indexes at different time points, unadjusted and adjusted for gender and age.

Predictive values of the candidate biomarkers were investigated through receiver operating characteristic (ROC) curves. Based on these curves, cut-off values for relapse prediction were assessed for each potential biomarker. Statistical significance was set at *p* < 0.05.

Figures show means ± SEM. The Statistical Package for Social Sciences (SPSS/Windows version 15.0; SPSS Inc., Chicago, IL, USA) and the software program GraphPad Prism (5.0 version; GraphPad, La Jolla, CA, USA) were used to perform statistical analyses.

## 3. Results

Seventy-nine patients were selected for the study, of which two were not eligible for further analysis because of death exitus before 72 h from admission. Thirty-six patients (71%) developed sepsis during admission, and there was clinical suspicion at 4.2 ± 1.3 days after admission. The clinical and demographic characteristics of patients are shown in Table 1. The severity indicators showed a high APACHE II score of 21 (SD: 7) and a SOFA score of 7 (SD: 4). According to the sepsis stratification, patients who developed sepsis during the follow-up had higher basal APACHE II and SOFA scores (APACHE II score—septic patients: 22 ± 6, non-septic patients: 19 ± 9; *p* = 0.05; SOFA score—septic patients: 8 ± 3, non-septic patients: 4 ± 3; *p* < 0.001). Both APACHE II and baseline SOFA were calculated immediately after patient inclusion. Patients who developed sepsis spent more time in the ICU than those that did not develop sepsis during the follow-up (septic patients: 25 ± 15 days; non-septic patients: 10 ± 7 days; *p* < 0.001). In addition, mechanical ventilation was required for longer durations in cases of septic patients (septic patients: 17 ± 13 days, non-septic patients: 4 ± 8 days; *p* < 0.001). There were no differences due to gender, age, and comorbidities or in mortality between groups. Thirty-six patients (71%) from the septic group showed positive blood cultures. Nine out of 55 septic patients developed septic shock during follow-up (Table 1).

C-reactive protein (CRP) was at higher levels on day +3 after admission (161.83 ± 133.42 mg/mL) than at the basal time point (71.90 ± 76.71 mg/mL) in all patients (Figure 2a). In contrast, the lymphocyte count, _M_HLA-DR rate, MFI of _M_HLA-DR, HLA-DR index, and sepsis index did not show any differences during the follow-up.

As shown in Figure 2, the dynamics of CRP (Figure 2b), sepsis index (Figure 2c), and _M_HLA-DR expression (Figure 2d) over time differed between groups. Septic patients showed increased levels of CRP (septic patients: 182.9 ± 132.9 mg/mL; non-septic patients: 93.46 ± 117.6 mg/mL; *p* = 0.030) (Figure 2b) and sepsis index (septic patients: 0.19 ± 0.19; non-septic patients: 0.08 ± 0.08; *p* = 0.010) (Figure 2c) on day +3 after admission. In contrast, the _M_HLA-DR rate was found to be decreased (septic patients: 81.7 ± 16.22%; non-septic patients: 88.53 ± 12.13%; *p* = 0.040) (Figure 2d). The _M_HLA-DR rate slowly recovered before 6 days, while the sepsis index remained higher in septic patients up to day +9. There were no differences in the lymphocyte count, the MFI of _M_HLA-DR, or the HLA-DR index between groups during the follow-up (Appendix A).

The HLA-DR monocyte expression was evaluated in ICU-admitted patients without infection and monitored for 15 days after admission. Considering the time of the sepsis diagnosis in the analysis, the _M_HLA-DR monocyte expression rate in septic patients significantly varied over time before the sepsis diagnosis (*p* = 0.001). As shown in Figure 3, the mean values of HLA-DR expression on the monocyte surface, measured as MFI HLA-DR on monocytes, were found to be decreased in septic patients and recovered after infection. The mixed model–interaction test showed that septic patients showed lower significant differences over time in the _M_HLA-DR monocyte expression, and the differences remained significant after multivariate adjustments for gender, age, and mechanical ventilation (*p* < 0.019). In addition, the percentage of HLA-DR^+^ monocytes tended to differ in septic patients over time (*p* = 0.09).

Patients who presented high sepsis indexes showed a higher risk of developing sepsis (OR: 2.71, *p* < 0.002). Moreover, statistical differences were found in the sepsis index means over time in septic patients vs. non-septic patients (*p* = 0.005).

As seen in the ROC curve analysis, CRP levels were significant predictors of the development of sepsis, followed by the _M_HLA-DR expression rate and MFI of _M_HLA-DR, while the lymphocyte count (*p* = 0.286) and sepsis index (*p* = 0.05) were not significant predictors. The area-under-the-curve values (AUCs) were the highest for CRP levels (AUC of 0.765, *p* < 0.001), followed by the _M_HLA-DR expression rate (AUC of 0.666; *p* < 0.001) and MFI of _M_HLA-DR (AUC of 0.654; *p* < 0.001). Figure 4 shows boxplots of CRP levels and _M_HLA-DR expression in septic and non-septic patients. To predict the development of sepsis, optimal cut-offs were CRP level > 106.90 mg/mL (74.19% sensitivity, 69.49% specificity), _M_HLA-DR expression rate < 72.80% (45.31% sensitivity, 89.47% specificity), and an MFI of _M_HLA-DR < 1882 (73.53% sensitivity, 53.76% specificity).

## 4. Discussion

The aim of this study was to identify biomarkers of immunoparalysis to predict which patients have an increased risk of developing sepsis at the ICU. The candidate biomarkers evaluated were (i) the HLA-DR expression rate on monocytes, (ii) the mean fluorescence intensity of HLA-DR on monocytes, (iii) the HLA-DR index, and (iv) the sepsis index. Our results showed that a pro-inflammatory/anti-inflammatory imbalance before infection produces an increased risk of developing sepsis in critical neurologic patients, and this risk is increased in patients who remain hospitalized for long periods.

The initial injury that leads to their admission to the ICU may trigger an imbalance between the processes of inflammation and immunosuppression. These processes involve the activation of several intracellular pathways, resulting in the production of pro-inflammatory cytokines. In parallel, a Compensatory Anti-inflammatory Response Syndrome (CARS) is activated as a temporary protective effect during the first hours after the injury. If this immunosuppressive state is maintained over time, it can produce immunoparalysis, predisposing the patient to infectious complications and the development of sepsis [7].

The decrease in the _M_HLA-DR on circulating monocytes has been mostly accepted as a reliable marker of immunoparalysis in septic patients [20,21]. This molecule reflects the loss of monocytes’ ability to present antigens and, consequently, to activate lymphocytes. Different authors studied the association between expression levels of the _M_HLA-DR and the prediction of sepsis [11,13,20]. However, the results obtained in those studies were not conclusive [13,22,23]. Differences in study design, such as monitoring of time points, as well as the lack of standardization of flow cytometry protocols, might be, in part, responsible for the discordant results. To avoid this variability due to the technical procedure, we used a standardized flow cytometry methodology, which can be easily transferred to other centers.

A prospective study in trauma patients without infection admitted to the ICU showed that the _M_HLA-DR expression was decreased in patients who developed sepsis during the follow-up [20]. Similar results were found in our study—involving patients with severe neurological injury—where the _M_HLA-DR on day +3 of follow-up was lower in patients that developed sepsis at later time points. To our knowledge, no other longitudinal and predictive studies have been performed analyzing _M_HLA-DR expression in ICU patients. These results collectively indicate that the analysis of the _M_HLA-DR expression is a useful marker to monitor the immunocompetent status of the patients and assess their susceptibility to the development of infections.

Regarding CRP levels, we observed higher values in patients who developed sepsis on day +3 of follow-up. At the same time point, the sepsis index value, which provides information regarding the pro-inflammatory and anti-inflammatory balance, was elevated, indicating that patients that developed sepsis suffered an imbalance due to an increase of inflammatory mediators and a decrease of HLA-DR molecules. There was clinical suspicion 4 days after admission.

The early diagnosis of the septic process is important to establish an adequate therapeutic strategy and increase survival rates [24]. Currently, the markers used in clinical practice to support the diagnosis of sepsis are CRP and procalcitonin (PCT), but they have limitations. While CRP is highly sensitive, it lacks specificity for the diagnosis of sepsis; conversely, PCT is more specific but lacks sensitivity. Therefore, there is a need to look for biomarkers capable of providing a reliable diagnosis [25]. In this context, the analysis of the evaluated biomarkers (_M_HLA-DR expression and sepsis index) showed changes before the diagnosis of sepsis. In contrast, no differences were found in those patients that did not develop sepsis during follow-up. The ROC analysis showed that the _M_HLA-DR expression rate was the biomarker with the highest specificity, and CRP was the one with the highest sensitivity to predict sepsis.

Taking all the results mentioned above into consideration, we propose a diagnostic algorithm that can be implemented in the monitoring of critical neurological patients admitted to the ICU. First, immune-monitoring the sepsis index, CRP, and _M_HLA-DR expression rate biomarkers at admission and on day +3 allows an early predictive stratification of susceptible sepsis patients. In this context, our preliminary results showed that the optimal cut-off values can be >106.90 mg/mL for CRP and <72.80% for the _M_HLA-DR expression rate. Patients who fulfill both criteria should be classified as potentially liable to develop sepsis.

The present study has a number of limitations. First, it is a single-center study, and the findings need to be confirmed in a larger and independent cohort. Moreover, we have only analyzed a specific group of patients (patients with severe neurological injury). The applicability of these biomarkers should be tested in different pathological contexts, such as severe acute pancreatitis, trauma, burn, or surgery.

We found three potential biomarkers (_M_HLA-DR expression together with CRP and sepsis index) in peripheral blood able to classify ICU patients with a high risk of developing sepsis. The combination of biomarkers can be useful to stratify the risk of developing sepsis during hospitalization. These results have potentially important implications in the hospital care area, as the combination of parameters studied can facilitate the management of critically ill patients.

## Figures and Tables

**Figure 1 biomedicines-11-01836-f001:**
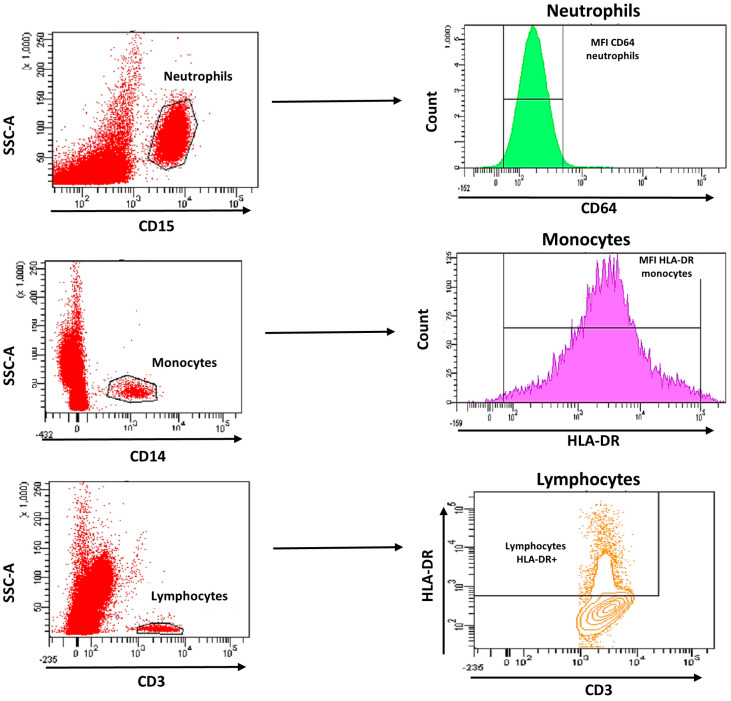
Gating strategy analysis using flow cytometry of CD64 expression on neutrophils and HLA-DR expression on monocytes and lymphocytes.

**Figure 2 biomedicines-11-01836-f002:**
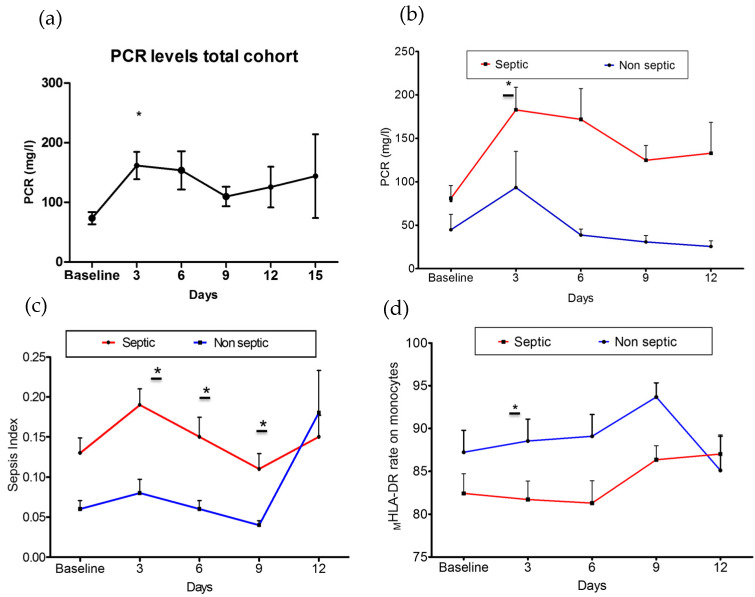
**Changes in measures of immune dysfunction based on subsequent sepsis status.** (**a**) Change in CRP levels of total patients included over time (*p* < 0.05). (**b**) Changes in CRP levels among groups (*p* = 0.03). (**c**) Change in sepsis index values among groups (*p* = 0.01). (**d**) Change in expression rate of _M_HLA-DR on monocytes among groups (*p* = 0.04). Septic group: Baseline: 55; day + 3: 55; day + 6: 48; day + 9: 45; day + 12: 39; day + 15: 24. Non-septic group: Baseline: 22; day + 3: 22; day + 6: 22; day + 9: 13; day + 12: 9; day + 15: 1. * *p* < 0.05.

**Figure 3 biomedicines-11-01836-f003:**
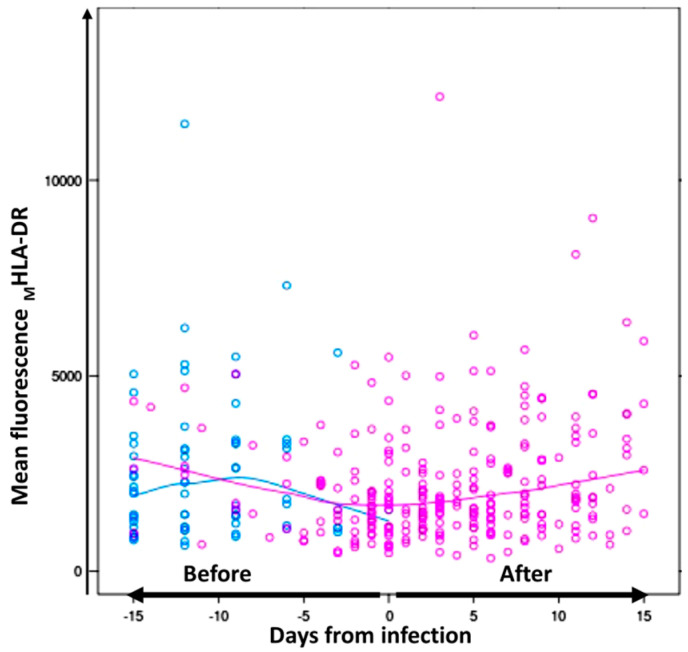
**Differences in the mean fluorescence of _M_HLA-DR on monocytes from septic and non-septic patients before infection.** Dynamic values of mean fluorescence intensity of _M_HLA-DR on monocytes from septic (**pink**) and non-septic (**blue**) patients during 15 days of follow-up over time. In the septic group, day 0 was considered when sepsis was diagnosed. The continuous lines represent the **mean HLA-DR fluorescence intensity** values in monocytes for septic patients (**pink**) and for non-septic patients (**blue**) before and after infection.

**Figure 4 biomedicines-11-01836-f004:**
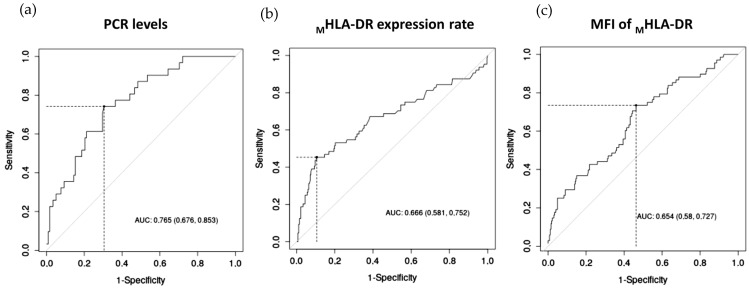
(**a**) ROC curves of CRP levels; (**b**) the expression rate of HLA-DR on monocytes; and (**c**) MFI of _M_HLA-DR for the diagnosis of patients with higher risk to develop sepsis during their stay in ICU.

**Table 1 biomedicines-11-01836-t001:** Demographic and clinical characteristics of patients.

Characteristics	Total Cohort *n* = 77	Septic *n* = 55	Non-Septic *n* = 22	*p*-Value
Female sex (n. of patients, (%))	26 (34)	17 (31)	9 (41)	0.400
Age (years), (IQR)	54 ± 16	54 ± 16	56 ± 16	0.520
Basal SOFA * score (IQR)	7 ± 4	8 ± 3	4 ± 3	**<0.001**
APACHE ** II score (IQR)	21 ± 7	22 ± 6	19 ± 9	0.050
Median hospital days (IQR)	21 ± 15	25 ± 15	10 ± 7	**<0.001**
Mechanic ventilation days (IQR)	14 ± 13	17 ± 13	4 ± 8	**<0.001**
**Comorbidities** **(n. of patients, (%))**				
COPD ***	6 (8)	4 (7)	2 (10)	0.790
Smoker	25 (32)	16 (29)	9 (41)	0.320
Alcoholism	15 (19)	9 (16)	6 (27)	0.270
Cardiopathy	8 (10)	6 (11)	2 (10)	0.810
Chronic kidney disease	7 (9)	5 (9)	2 (10)	1.000
Cirrhosis	2 (3)	2 (4)	0 (0)	0.360
**Exitus (n.patients, (%))**	14 (18)	11 (20)	3 (14)	0.510
**Blood culture** (n. of patients, (%))	36 (48)	36 (66)	NA	
**Adequate antibiotic treatment** (n. of patients, (%))	41 (53)	41 (75)	NA	
**Clinical suspicion sepsis days** (IQR)	4 ± 1	4 ± 1		
**Shock septic** (n. of patients, (%))	9 (12)	9 (16)	NA	

* SOFA = Sepsis-related Organ Failure Assessment scale, ** APACHE = Acute Physiology and Chronic Health Evaluation, *** COPD = Chronic obstructive pulmonary disease.

## Data Availability

Data sharing is not applicable to this article.

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
