# Peer review of "HLA-DR Expression on Monocytes and Sepsis Index Are Useful in Predicting Sepsis"

_biomedicines, 2023, doi:10.3390/biomedicines11071836_

Round 1

Reviewer 1 Report

In this work, authors evaluated the HLA-DR expression on the surface of monocytes (MHLA-DR ratio), the sepsis index (CD64 expression on neutrophils/MHLA-DR ratio) and C-reactive protein (CRP) with the development of sepsis. They prospectively enrolled 77 critically ill patients, 59 with stroke and 18 with traumatic brain injury. The biomarkers were tested at baseline and 3, 6, 9, 12 and 15 days later. Most patients (71%) developed sepsis. They concluded that periodic monitoring of the MHLA-DR expression together with CRP and sepsis index may help to identify patients at increased risk to develop sepsis. Authors performed flow cytometry analysis, plasma analysis and statistical analysis.

Subject definitions and methods are optimal. Figures and references are adequate.

I have found this paper relevant to the field of this journal. I have only but many minor comments.

Minor points:

1. Introduction:

You have concentrated only on the biomarkers which you had measured. However people have tried to find also others with association to sepsis, septic shock and mortality. For instance the publication: Cejkova P et al, Monitoring of the course of sepsis in hematooncological patients by extrapituitary prolactin expression in peripheral blood monocytes. Physiological Research, 2012: 61(5): 481-488. doi: 10.33549/physiolres.932262, etc. You should mention them.

2. Materials and Methods:

The line 128, remove the comma after the words “HLA-DR index“, and put the small letter for “index”. Similarly, put the small letter for “index” in the lines: 130 and 133.

3. Results:

1) The line 167, remove the comma after the words “HLA-DR index“.

2) The line 178, remove the comma after the parenthesis “(Fig 2 c) “.

3) The line 192, remove “a” after the words “Figure 3“.

4) The titles of Figure 1 (Gating strategy analysis by flow cytometry of CD64 expression on neutrophils 119 and HLA-DR expression on monocytes and lymphocytes.), Table 1 (Demographic and clinical characteristics of patients.) and Figure 4 (ROC curves for), use the bold letters (like in the titles of Figure 2 and Figure 3).

5) The text of Figure 2, change “(c) Sepsis index values for differences between septic and non-septic patients” into “(c) Sepsis index values for differences between groups” (line 175-176). For (b), (c) and (d), put exact p value: p = 0.030, p = 0.010 and p = 0.040, respectively.

6) Figure 2a, 2b, change “PCR” into “CRP” (in the title and on the y axes)!!! On the y axes, change “mg/ml” into “mg/mL”.

7) Similarly, Figure 4a, change “PCR” into “CRP” (in the title)!!!

8) Similarly, Supplementary Tables 1 and 2, change “PCR (mg/ml)” into “CRP (mg/mL)”!!!

4. Discussion:

1) The line 225, remove the comma after the words “HLA-DR index“, and put the small letter for “index”.

I recommend this paper for acceptation after minor revision in the journal.

 Minor editing of English language required.

Author Response

  1. Introduction:

We have added a commentary on biomarkers that are associated with better prognosis in patients with sepsis and added the suggested citation.

  1. Materials and Methods:

The line 128, remove the comma after the words “HLA-DR index“, and put the small letter for “index”. Similarly, put the small letter for “index” in the lines: 130 and 133. Done

  1. Results:

1) The line 167, remove the comma after the words “HLA-DR index“. Done

2) The line 178, remove the comma after the parenthesis “(Fig 2 c) “. Done

3) The line 192, remove “a” after the words “Figure 3“. Done

4) The titles of Figure 1 (Gating strategy analysis by flow cytometry of CD64 expression on neutrophils 119 and HLA-DR expression on monocytes and lymphocytes.), Table 1 (Demographic and clinical characteristics of patients.) and Figure 4 (ROC curves for), use the bold letters (like in the titles of Figure 2 and Figure 3). Done

5) The text of Figure 2, change “(c) Sepsis index values for differences between septic and non-septic patients” into “(c) Sepsis index values for differences between groups” (line 175-176). For (b), (c) and (d), put exact p value: p = 0.030, p = 0.010 and p = 0.040, respectively. Done

6) Figure 2a, 2b, change “PCR” into “CRP” (in the title and on the y axes)!!! On the y axes, change “mg/ml” into “mg/mL”. Done

7) Similarly, Figure 4a, change “PCR” into “CRP” (in the title)!!! Done

8) Similarly, Supplementary Tables 1 and 2, change “PCR (mg/ml)” into “CRP (mg/mL)”!!!. Done

  1. Discussion:

1) The line 225, remove the comma after the words “HLA-DR index“, and put the small letter for “index”. Done

Reviewer 2 Report

Thank you very much for contributing this interesting paper. 

However, I have some considerations to be made: 

Methods:

-      What is the difference between expression rate of HLA-Dr and MIF HLA-DR. How can you measure MIF standardized as the fluorescence intensity of each run depends on the laser and can differ between runs?

-      Was is defined as baseline in the septic patients’ groups?

Results:

-      Was the basal SOFS score of the septic patients measured after study inclusion or at the time point if sepsis onset?

-      Please indicate the time until sepsis development in table 1.

-      The percentages in table 1 seem to be inconsistent.

-      Why are there no values for Blood culture (positive blood culture) and adequate antibiotic treatment in the sepsis group.

-      Figure 2: do you mean PCR kevels in figure 2a or is it CRP?

-      Please write n-counts for each time point.

-      I do not understand the meaning of figure 3. Perhaps it would be interesting to show the development of each single patient?

-      In addition, HLA-DR molecules per monocyte is the gold standard for HLA measurement in septic patients (e.g. Identification of a sub-group of critically ill patients with high risk of intensive care unit-acquired infections and poor clinical course using a transcriptomic score). Why didn’t you measure this? Again, what is the meaning of MIF HLA-DR?

-      In the conclusion you state that you “found a combination of biomarkers (MHLA-DR expression together with CRP and sepsis index) in peripheral blood able to stratify ICU patients with high risk to develop sepsis.”. However, in your ROC analysis you only show one marker. Could you make a combination of the markers? Perhaps you can use AI to identify clusters in your patients’ groups?

The english seems to be fine. Some spelling mistakes have ro be deteleted. 

The formating in figure 4 and table one should be mofied and carefully corrected. 

Author Response

1. Methods:

-      What is the difference between expression rate of HLA-Dr and MIF HLA-DR. How can you measure MIF standardized as the fluorescence intensity of each run depends on the laser and can differ between runs?

-      Was is defined as baseline in the septic patients’ groups?

Thank you for your comments, in the methodology section we added a paragraph including the definition and differences of HLA-DR rate and MFI.  

HLA-DR rate is the number of monocytes that express HLA-DR molecules on surface. As the HLA-DR is a molecule that can be expressed to a greater or lesser quantity depending on the functionality of the cell, we also measure HLADR mean fluorescence intensity (MFI) to quantify it. 

In the methodology section we have described the calibration process to ensure that the laser voltages of the cytometer are adjusted to it. For this we use beads with known MFI and adjust the lasers in each independent test.   

2. Results:

-      Was the basal SOFS score of the septic patients measured after study inclusion or at the time point if sepsis onset?

SOFA score was measured after study inclusion. We have added a comment to clarify this. 

-      Please indicate the time until sepsis development in table 1.

We have added this information in the table.

-      The percentages in table 1 seem to be inconsistent.

We have corrected them. 

-      Why are there no values for Blood culture (positive blood culture) and adequate antibiotic treatment in the sepsis group.

We have added them.

-      Figure 2: do you mean PCR kevels in figure 2a or is it CRP?

It is CRP, we have corrected the error in figure 2. 

-      Please write n-counts for each time point.

We have added them. 

-      I do not understand the meaning of figure 3. Perhaps it would be interesting to show the development of each single patient?

We have added information for a better understanding of figure 3. 

-      In addition, HLA-DR molecules per monocyte is the gold standard for HLA measurement in septic patients (e.g. Identification of a sub-group of critically ill patients with high risk of intensive care unit-acquired infections and poor clinical course using a transcriptomic score). Why didn’t you measure this? Again, what is the meaning of MIF HLA-DR?

In our experimental design we have used a combination of different fluorochromes to label HLA-DR and CD 64 with respect to the Quantibrite reagent. We use PE to label CD 64 which is a non-constitutive marker in neutrophils. We are interested in analyzing the expression of CD 64 in neutrophils to detect early infection. Being marked with PE, we could know in the future the number of molecules on the surface by their 1:1 ratio of fluorochrome: monoclonal antibodies. On the other hand, for HLA-DR we are interested in monitoring the change in its expression, decrease during follow-up, to detect immunosuppression. Our approach allows us to combine both studies.  

-      In the conclusion you state that you “found a combination of biomarkers (MHLA-DR expression together with CRP and sepsis index) in peripheral blood able to stratify ICU patients with high risk to develop sepsis.”. However, in your ROC analysis you only show one marker. Could you make a combination of the markers? Perhaps you can use AI to identify clusters in your patients’ groups?

We are currently recruiting more patients to perform a predictive model. Our objective is to obtain a sepsis risk score to classify patients with immunoparalysis. In addition, it will allow us to validate results tha we present.  

Round 2

Reviewer 2 Report

Thank you very much for changing the manuscript. All my suggestions were properly added to the paper. 

English seems to be fine. However I suggest to make a final spell check.